# Interlayer epitaxy of wafer-scale high-quality uniform AB-stacked bilayer graphene films on liquid Pt₃Si/solid Pt

Wei Ma [1,2], Mao-Lin Chen[1,2], Lichang Yin[1], Zhibo Liu[1], Hui Li[1,2], Chuan Xu[1], Xing Xin[1,3], Dong-Ming Sun [1,2], Hui-Ming Cheng [1,2,4] & Wencai Ren [1,2]

Large-area high-quality AB-stacked bilayer graphene films are highly desired for the applications in electronics, photonics and spintronics. However, the existing growth methods can only produce discontinuous bilayer graphene with variable stacking orders because of the non-uniform surface and strong potential field of the solid substrates used. Here we report the growth of wafer-scale continuous uniform AB-stacked bilayer graphene films on a liquid Pt₃Si/solid Pt substrate by chemical vapor deposition. The films show quality, mechanical and electrical properties comparable to the mechanically exfoliated samples. Growth mechanism studies show that the second layer is grown underneath the first layer by precipitation of carbon atoms from the solid Pt, and the small energy requirements for the movements of graphene nucleus on the liquid Pt₃Si enables the interlayer epitaxy to form energy-favorable AB stacking. This interlayer epitaxy also allows the growth of ABA-stacked trilayer graphene and is applicable to other liquid/solid substrates.

[1] Shenyang National Laboratory for Materials Science, Institute of Metal Research, Chinese Academy of Sciences, 110016 Shenyang, P. R. China. [2] School of Material Science and Engineering, University of Science and Technology of China, 110016 Shenyang, P. R. China. [3] University of Chinese Academy of Sciences, 110016 Shenyang, P. R. China. [4] Shenzhen Geim Graphene Center, Tsinghua-Berkeley Shenzhen Institute (TBSI), Tsinghua University, 1001 Xueyuan Road, 518055 Shenzhen, P. R. China. Correspondence and requests for materials should be addressed to W.R. (email: wcren@imr.ac.cn)

The widely tunable bandgap of AB-stacked bilayer graphene (AB-BLG) leads to many interesting physical phenomena and makes it a promising material for a number of electronic, photonic and spintronic devices[1–6]. For these applications, it is critical to develop a scalable approach to synthesize large-area high-quality uniform AB-BLG films that can be manipulated to complex devices and integrated in silicon device flow. Mechanical exfoliation of highly ordered graphite provides a direct way to obtain high-quality AB-BLG, but suffers from low yield, small size (typically smaller than tens of micrometers), and poor controllability on the thickness[1–8]. In recent years, chemical vapor deposition (CVD) has emerged as a promising scalable method for producing AB-BLG by using solid metal foils as substrates such as Cu and Cu–Ni alloy[9–15]. However, the weak van der Waals potential between graphene layers cannot completely screen the strong potential field of solid substrates[16]. In this case, the nucleation, growth and orientation of the second layer in BLG are more strongly influenced by the solid substrates than the first layer. On the other hand, solid metal substrates usually have variable crystallographic lattices, defects, big surface roughness and abundant grain boundaries[10,12,13]. Such non-uniform surface structures and strong potential field of solid substrates result in only discontinuous BLG films with variable stacking orders and small grain size[9–15] (Supplementary Table 1). Although BLG films with half-millimeter-sized grains have been synthesized by oxygen-activated CVD process using a Cu pocket substrate, the coverage of BLG is only ~75% with the AB-stacked area around 80%[13].

Here, we show that a liquid top layer can screen the strong potential of the solid substrate underneath to enable interlayer epitaxy of AB-BLG, in which the second layer grows epitaxially with the first layer graphene as the seed layer. Using a shell-core structured liquid $Pt_3Si$/solid Pt substrate, we grow high-quality wafer-scale 100% AB-BLG films with 100% coverage and millimeter-sized grains by CVD. The Young's modulus and breaking strength of the films are as high as 1.02 TPa and 122 GPa, and the mobility based on large-size field effect transistors (FETs) can reach 2100 $cm^2 V^{-1} s^{-1}$ at room temperature with a tunable bandgap >26 meV at displacement field of 1.0 V $nm^{-1}$. The interlayer epitaxy mechanism is analyzed based on isotope experiments and density functional theory (DFT) calculations. It is found that the 2nd layer is grown underneath the first layer by precipitation of carbon atoms from the solid Pt core with high carbon solubility through the liquid $Pt_3Si$ shell with low carbon solubility, and the small energy requirement for the movement of graphene nucleus on liquid $Pt_3Si$ enables the interlayer epitaxy to form energy favorable AB stacking. We also demonstrate the growth of ABA-stacked trilayer graphene by using the same substrate and AB-BLG films by using a similarly structured liquid Pd silicide/solid Pd substrate.

## Results

**Interlayer epitaxy of AB-BLG films on $Pt_3Si$/Pt substrate**. The $Pt_3Si$/Pt substrate was synthesized by coating a 500-nm-thick Si film on a 250-μm-thick polycrystalline Pt foil by magnetron sputtering, followed by annealing in $H_2$ atmosphere at 1095 °C for 12 h. According to the Pt-Si phase diagram[17], Si element can form stoichiometric intermetallic compound, platinum silicide, with Pt element. As shown in the energy dispersive X-ray spectroscopy (EDS) maps in Fig. 1a, b, the Si element is homogenously distributed on the surface and grain boundaries of the Pt foil, forming a shell-core structure of platinum silicide encapsulating Pt grains. Further X-ray diffraction (XRD) analysis reveals that the platinum silicide is $Pt_3Si$ (Fig. 1c)[18].

To grow AB-BLG, as shown in Supplementary Fig. 1, the $Pt_3Si$/Pt substrate was first kept in a mixed gas flow of methane ($CH_4$) and hydrogen ($H_2$) at 1100 °C for a short time, and then slowly decreased to 1025 °C with a constant cooling rate (12.5 °C $h^{-1}$), followed by rapid cooling to room temperature. Since the melting point of $Pt_3Si$ is around 830 °C[17], the $Pt_3Si$ regions in the substrate turned into a liquid phase during the growth process, forming a liquid $Pt_3Si$/solid Pt shell-core structure. In this structure, the solid Pt core with high carbon solubility (0.86 wt.% at 1100 °C[17]) acts as a reservoir of carbon atoms, while the liquid $Pt_3Si$ shell with low carbon solubility provides a diffusion channel of carbon atoms (Supplementary Fig. 2). Furthermore, the liquid $Pt_3Si$ surface can screen the strong potential field of solid Pt and eliminate the influence of variable crystallographic lattice, defects, surface roughness and grain boundaries that are present in the Pt foil, which is beneficial for the growth of uniform graphene.

As shown in Fig. 1d, uniform monolayer graphene (MLG) was formed in 10 min at 1100 °C. Meanwhile, the excess carbon atoms diffused through the liquid $Pt_3Si$ shell into the solid Pt core. In the following slow-cooling process, the carbon solubility in the Pt core was decreased with decreasing the temperature[17], leading to the precipitation of carbon atoms and consequently the formation of the second layer domains (Fig. 1e). With prolonging the growth time, these domains expanded and gradually joined together, and eventually a very uniform wafer-scale (~1.5 × 3.5 $cm^2$) continuous BLG film was formed (Fig. 1f, g). The film shows an optical transmittance of 95.42 ± 0.08% at 550 nm (Fig. 1h) and two lattice fringes at the folded edges (Fig. 1i), confirming that it is bilayer[19]. The size of the BLG films can be easily scaled up by using larger substrates. Moreover, no additional layers were formed even by further extending the slow-cooling time (120 min), suggesting the robustness of our method for BLG growth. In contrast, under the same experimental conditions, only non-uniform multilayers and uniform MLG were synthesized on Pt and $Pt_3Si$, respectively (Supplementary Fig. 3). For structure characterizations and property measurements, the CVD-grown BLG graphene films were transferred onto the $SiO_2$/Si substrate or TEM grid by electrochemical bubbling[20]. As shown in Supplementary Fig. 4, the substrate remains its original structure after transfer and can be repeatedly used for BLG growth.

**Structural characterizations of AB-BLG films**. We first used Raman spectroscopy to characterize the stacking order, quality and uniformity of the BLG films. As shown in Fig. 2a, our BLG film shows the same Raman features as the exfoliated AB-BLG. The asymmetrical 2D peak has a full width at half maximum (FWHM) of ~54 $cm^{-1}$, about twice that of MLG, can be fitted into four Lorentzian bands with different frequencies (Fig. 2b), and shows a lower intensity than the G peak with $I_{2D}/I_G$ of ~0.77. All these features are also consistent with those of reported exfoliated AB-BLG[21]. The absence of D peak suggests the high quality of our samples. We also performed Raman mapping on a randomly selected 600 × 550 $μm^2$ region with a laser spot size of 1 μm and step of 5 μm. The very uniform $I_{2D}/I_G$ and 2D peak FWHM maps indicate the high uniformity of our AB-BLG film (Fig. 2c, d). Further analyses on the 13,431 Raman spectra indicate that the $I_{2D}/I_G$ and 2D peak FWHM are narrowly distributed in the range of 0.51–0.83 and 52.4–57.1 $cm^{-1}$, respectively (Fig. 2c–e). It is worth noting that these values are well consistent with those of mechanically exfoliated AB-BLG (Fig. 2e and Supplementary Fig. 5), confirming the perfect AB stack of our CVD-grown BLG film. In sharp contrast, for the reported CVD-grown AB-BLG[9,10,12,14,15], the $I_{2D}/I_G$ and 2D peak FWHM have much larger variation and are located between the values of exfoliated AB-BLG and MLG (Fig. 2e and Supplementary Fig. 5).

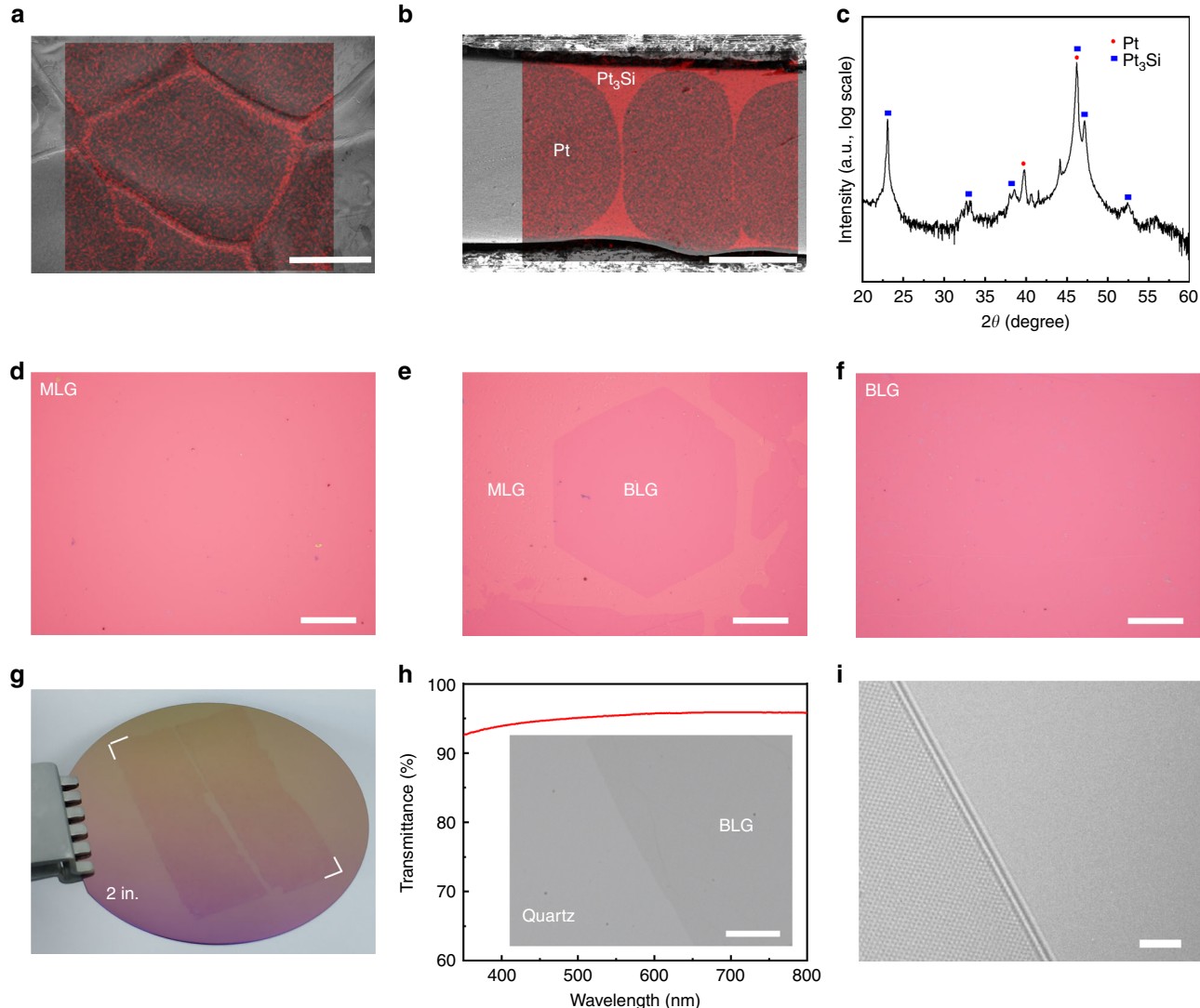

**Fig. 1** Interlayer epitaxy of wafer-scale AB-BLG films by CVD on shell-core structured liquid $Pt_3Si$/solid Pt substrate. **a**, **b** EDS maps of the Si $K_{\alpha 1}$ peak overlapped with the corresponding scanning electron microscopy (SEM) images of the surface (**a**) and cross section (**b**) of the substrate. Note that the Si element is distributed on the very surface and the grain boundaries of the Pt foil, forming a shell-core structure. **c** XRD pattern of the substrate (on log scale), showing the coexistence of Pt and $Pt_3Si$. **d**–**f** Optical images showing the formation process of BLG. **d** A monolayer graphene (MLG) film obtained after 10 min growth at 1100 °C. **e** A MLG film with the second layer domains obtained after slow cooling for 4 h. **f** A continuous BLG film obtained after slow cooling for 6 h. The cooling rate keeps 12.5 °C h$^{-1}$. **g** Photograph of two ~1.5 × 3.5 cm$^2$ CVD-grown AB-BLG films on a 2-inch SiO$_2$/Si wafer. **h** Transmittance spectra of a BLG film on a quartz substrate measured by UV-vis-NIR spectroscopy (measurement area: 0.5 × 0.5 cm$^2$). Inset shows a photograph of a quartz substrate partially covered by a BLG film. **i** High-resolution transmission electron microscopy (HRTEM) image of a folded edge of the BLG film, showing a well-defined interlayer spacing of ~0.34 nm. Scale bars: **a**, **b**, **d**, **e**, **f** 100 μm; **h** 10 μm; **i** 2 nm

We further characterized the atomic-level structure of the films using HRTEM, scanning transmission electron microscopy (STEM) and selective area electron diffraction (SAED). The following four features were observed. First, the film shows a hexagonal SAED pattern with $I_{\{1\bar{2}10\}}/I_{\{1\bar{1}10\}}$ of ~3 (Fig. 2f and Supplementary Fig. 6), characteristic of AB-BLG[8]. Second, atomic-resolution STEM image shows alternative bright and dark spots (Fig. 2g), both of which have hexagonal symmetry with a spacing of 0.25 nm, close to $\sqrt{3}a$. As reported[22], this is also a unique structure characteristic of AB-BLG, in which the bright and dark spots correspond to the overlapped AB atoms and unoverlapped A or B atoms, respectively. Third, large-area HRTEM image shows no Moiré pattern (Fig. 2h), which is a characteristic of twisted-BLG[23]. Fourth, no defects or disorders are observed in HRTEM and STEM images even in a large area. These observations give further evidence that our CVD-grown films are high-quality AB-BLG.

More surprisingly, we found that the two layers in BLG films follow epitaxial growth in our CVD process. Fig. 2f shows a typical dark-field TEM image of the grain boundary (GB) region in an AB-BLG film. It is very important to note that the top and bottom layers have overlapped GB. Moreover, the two sides of the GB are both AB-stacked. The corresponding SAED patterns indicate the two neighboring AB-BLG domains are misaligned with 12° difference in lattice orientation. These results give clear evidence that the lattice orientation of the second layer is determined by the pre-formed first layer, i.e. interlayer epitaxy. This is sharply contrast to the growth of BLG on solid metal foils, where the second layer is more strongly influenced by the metal substrate and the stacking order varies when the second layer

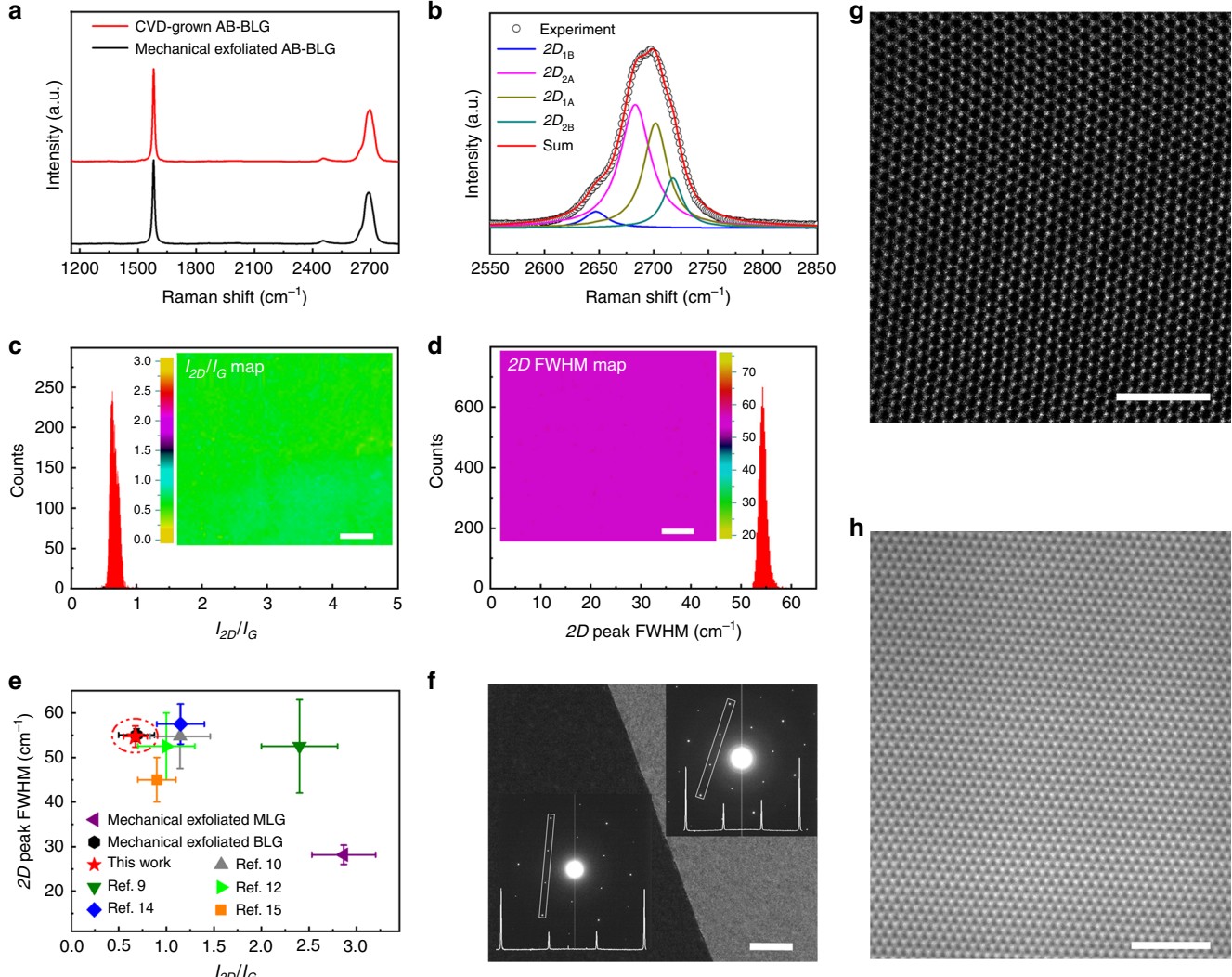

**Fig. 2** Structure characterization of the CVD-grown AB-BLG films by interlayer epitaxy. **a** Raman spectra of AB-BLG synthesized by CVD and mechanical exfoliation. **b** 2D peak of the CVD-grown BLG, which was fitted by four Lorentzian peaks. **c, d** Raman maps and statistical histograms of $I_{2D}/I_G$ (**c**) and 2D peak FWHM (**d**) of a CVD-grown AB-BLG film. Raman mapping was performed in a randomly selected region of $600 \times 550\ \mu m^2$ with a spot size of 1 μm and step of 5 μm. **e** Comparison of 2D peak FWHM vs $I_{2D}/I_G$ for the AB-BLG samples synthesized by our CVD method on liquid $Pt_3Si$/solid Pt substrate, the CVD on solid metal substrates[9,10,12,14,15], and mechanical exfoliation. The 2D peak FWHM and $I_{2D}/I_G$ of MLG synthesized by mechanical exfoliation were also presented. The error bars show the variation of the 2D peak FWHM and $I_{2D}/I_G$ for each sample. **f** Dark-field TEM image of the grain boundary region of an AB-BLG film. The inset shows SAED patterns taken from both sides of the GB. **g, h** Atomic-resolution STEM (**g**) and HRTEM (**h**) images of a AB-BLG film. Scale bars: Insets of **c, d** 100 μm; **f** 500 nm; **g, h** 2 nm

grows across the GB of the first layer[10,12,13]. This interlayer epitaxy allows us to tune the grain size and GBs of the AB-BLG film by controlling the grain structure of the first layer graphene. For instance, we synthesized MLG with millimeter-sized grains by using a small flow rate ratio of $CH_4$ to $H_2$ in the first step (Supplementary Fig. 7), which enables the growth of continuous AB-BLG film with millimeter-sized grains by interlayer epitaxy (Fig. 1f). In addition, the unique bilayer GBs provide a platform for exploring unique topological phases and valley physics in graphene[6].

**Mechanical and electrical properties of AB-BLG films.** We measured the mechanical properties of the AB-BLG films by using nanoindentation as reported previously[24,25]. To do this, the AB-BLG film was transferred onto a $SiO_2$/Si substrate with an array of holes with 1-μm diameter to create suspended membrane. Figure 3a shows typical force-displacement curves

obtained by using an atomic force microscopy (AFM) with a diamond tip of 11.07 nm radius (r). The identical loading and unloading curves demonstrate the elastic behavior of the film and no slippage at the periphery of the hole. Due to the slip between the two layers of BLG[25], the curves show a softening for displacement larger than ~50 nm. Therefore, we fitted the data up to 50 nm of displacement to obtain the second-order elastic modulus of the membranes[25]. Fitting the results of 23 individual tests yield a mean 2D Young's modulus ($E^{2D}$) of 684 N m$^{-1}$ (±50 N m$^{-1}$) (Fig. 3b). This value corresponds to a bulk Young's modulus of 1.02 TPa, assuming 0.670 nm as the thickness of AB-BLG[25]. We also performed nanoindentation measurements on the CVD-grown high-quality single-crystal MLG with the same method, which give an average 2D Young's modulus of ~334.02 N m$^{-1}$ and breaking strength of ~55.31 N m$^{-1}$ (Supplementary Fig. 8). These results are well consistent with those of reported MLG[26], confirming the reliability of the AFM nanoindentation

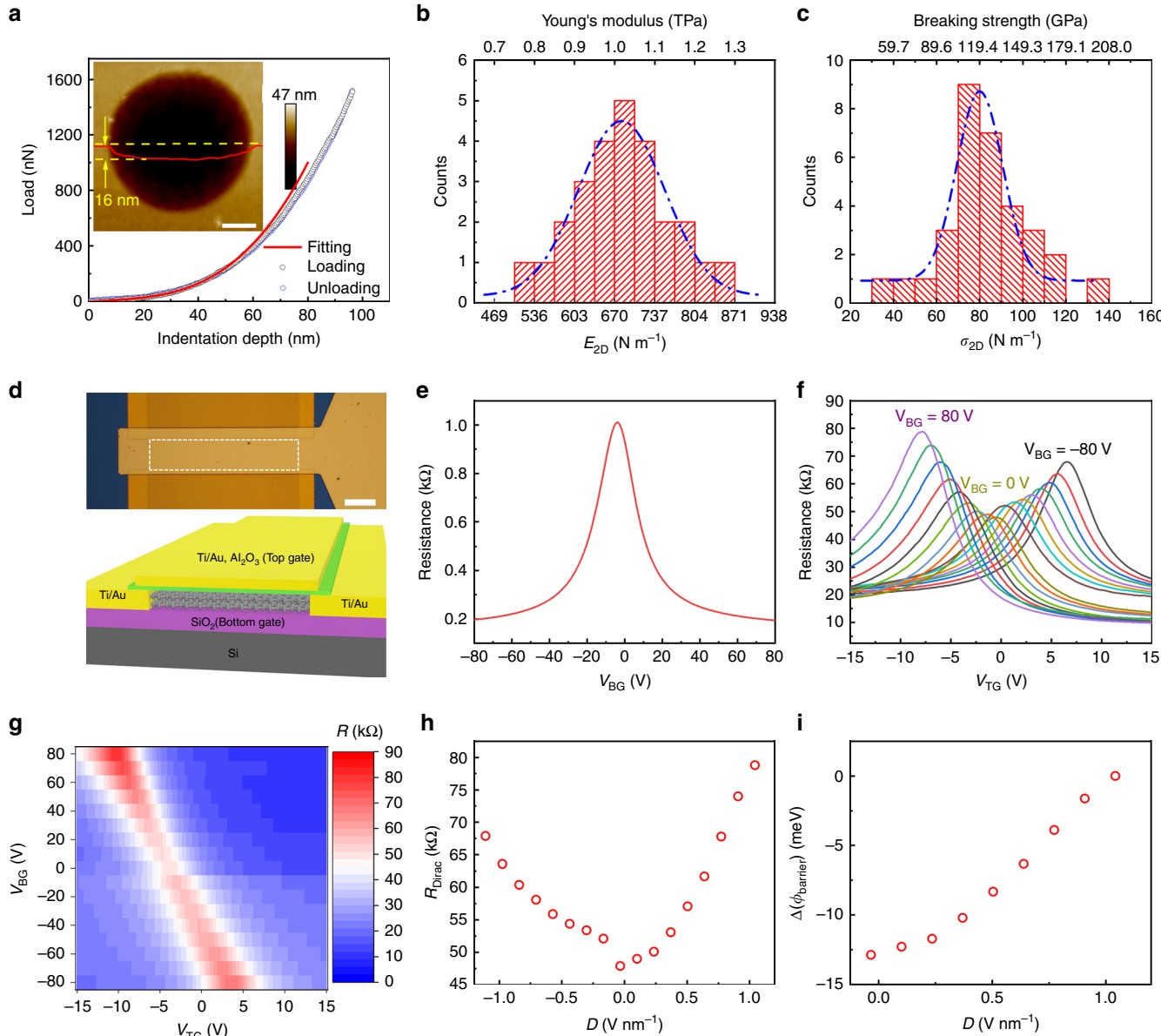

**Fig. 3** Mechanical and electrical properties of the CVD-grown AB-BLG films by interlayer epitaxy. **a** Typical force-displacement curves of an AB-BLG film in AFM nanoindentation. The red line is the fitting curve, and the inset is an AFM image of the suspended AB-BLG film before indentation test and the height profile (red line) along the dashed yellow line showing ~16 nm indentation in the hole. **b**, **c** Histograms of elastic stiffness (**b**) and breaking strength (**c**). Dashed lines represent Gaussian fits to the data. **d** Optical image (top) and illustration (bottom) of a dual-gate AB-BLG FET device. The dashed square in the optical image shows the AB-BLG channel (80 μm × 20 μm) underneath the top gate. The top-gate dielectric is 60 nm $Al_2O_3$, and the back-gate dielectric is 290 nm $SiO_2$. **e** Resistance versus $V_{BG}$ curve measured at room temperature. **f** The room temperature transfer characteristics of a dual-gate AB-BLG FET. $V_{BG}$ is varied from −80 to 80 V with step of 10 V. **g** Color plot of resistance as function of both $V_{TG}$ and $V_{BG}$ obtained from the data in (**f**). **h** Resistance at Dirac point under different electrical displacement, $D$. **i** Variation of Schottky barrier height, $\Delta(\phi_{barrier})$, as a function of $D$, inferred from the off currents at the charge neutrality point in (**f**). Scale bars: Inset of **a** 300 nm; **d** 20 μm

measurements in this work. Since the breaking strength of BLG varies approximately with $r^{1/2}$, based on the measured strength of high-quality CVD-grown MLG and the tip radius (11.07 nm) used in our experiments, the mean 2D strength ($\sigma^{2D}_m$) of AB-BLG was estimated to be 82 N m$^{-1}$ (Fig. 3c) using the method shown in ref. [25], corresponding to a bulk strength of 122 GPa. Note that both the Young' modulus and breaking strength of our samples are in good agreement with those (1.04 TPa and 126 GPa) of exfoliated AB-BLG[25], further confirming the high crystalline quality of our CVD-grown AB-BLG film. Because the indentation area (1 μm) is significantly smaller than the grain size (millimeter scale) of AB-BLG film, the influence of GBs on the

mechanical properties of AB-BLG film can be ignored in our experiments, but it is a very interesting topic deserving further studies in the future.

Large-size dual-gate FETs were fabricated and measured at room temperature to demonstrate the bandgap modulation of our AB-BLG films, a unique property of AB-BLG[1,2]. As shown in Fig. 3d, the AB-BLG film was used as channel material encapsulated between 290 nm $SiO_2$ layer and 60 nm $Al_2O_3$ layer, which act as back- and top-gate dielectrics, respectively. The channel size is 80 μm (width) × 20 μm (length), which is much larger than those reported, typically <10 μm × 10 μm. Compared to small-size device, large-size device has much higher

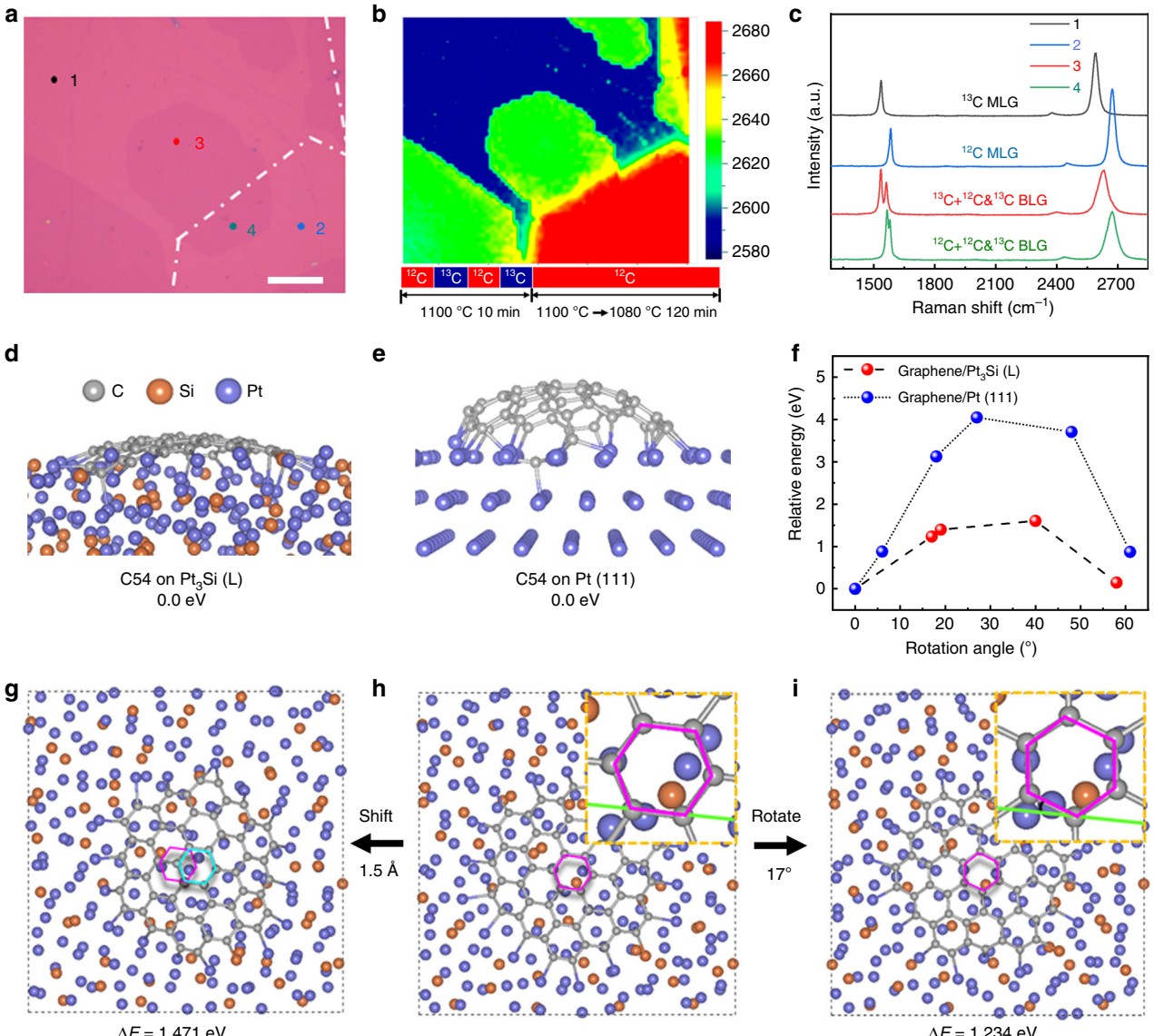

**Fig. 4** Interlayer epitaxial growth mechanism of AB-BLG films on liquid Pt₃Si/solid Pt substrate. **a, b** Optical image (**a**) and the corresponding isotopically labeled Raman map of *2D* peak position (**b**) of an AB-BLG transferred on SiO₂/Si substrate. The dashed lines in (**a**) indicate the boundary of $^{12}C$ and $^{13}C$ isotopic rings in the first layer. **c** Raman spectra of isotopically labeled monolayer and bilayer graphene regions indicated in (**a**). **d, e** The side view diagrams of the atomic structures of graphene nucleus (C54) on liquid Pt₃Si surface (**d**) and solid Pt (111) surface (**e**) at 1500 K. **f** The relative energies of graphene nucleus (C54) on solid Pt (111) and liquid Pt₃Si surfaces as a function of rotation angle at 1500 K. **g–i** The diagrams of three representative atomic structures of graphene nucleus (C54) on liquid Pt₃Si surface. The atomic structure in (**h**) can be converted into the atomic structures in (**g**) and (**i**) by a translation of 1.5 Å and 17° rotation, respectively. The corresponding energy requirements for each movement were shown below. Scale bar: **a** 50 μm

requirements on the quality and uniformity of the channel material since the probability of defects and cracks dramatically increases with the channel size, which can easily lead to device performance degradation[27]. The resistance $R$ versus the back-gate voltage ($V_{BG}$) plot shows typical ambipolar characteristics expected for graphene devices (Fig. 4e). The curve exhibits a narrow and symmetric resistivity peak, with the charge neutrality point $V_{BG} = -4$ V. These features suggest low densities of charged impurities and defects, and hence high crystalline quality. The derived carrier mobility is as high as ~2100 cm² V⁻¹ s⁻¹ at room temperature, which is comparable to those (1000–3000 cm² V⁻¹ s⁻¹) of exfoliated AB-BLG with similar device structures but much smaller channel size[1] (Supplementary Table 1). Importantly, as shown in Fig. 3f, when sweeping the top-gate voltage ($V_{TG}$) at different fixed $V_{BG}$, the

maximum resistance ($R_{Dirac}$) attained at the charge neutrality point increases with increasing $V_{BG}$ in both the positive and negative directions. Figure 3g, h clearly shows that $R_{Dirac}$ increases when increasing the displacement field ($D$), confirming the tunability of bandgap with perpendicular dipole electric field. We further extracted the lower limit of the bandgaps at each fixed $D$ based on the Schottky barrier height inferred from the off-current at each charge neutrality point according to the method shown in ref. [2] (Fig. 3i). A bandgap >26 meV was achieved at $D = 1.0$ V nm⁻¹, and the corresponding on/off ratio is ~10, which are comparable to that of exfoliated AB-BLG based on similar device structure[2]. It has been reported that h-BN is an ideal dielectric that is beneficial to maintain the intrinsic properties of graphene because it has atomically smooth surface that is relatively free of dangling bonds and charge traps, similar lattice

constant with graphene, large optical phonon modes and a large electrical bandgap[28]. We believe that the device performances of our AB-BLG films can be further improved by using h-BN flakes as back- and top-gate dielectrics as reported in ref. [13].

**Mechanism of interlayer epitaxial growth**. To understand the growth mechanism of AB-BLG films, we visualized the time evolution of graphene growth by alternating $^{12}CH_4$ and $^{13}CH_4$ for periods of 2.5 min in the first step to form a MLG film (in total 10 min), followed by using only $^{12}CH_4$ in the second step with a shorter cooling time to form isolated second layer domains (Fig. 4b), and then mapping the isotopic composition of the grown BLG by Raman spectroscopy. As shown in Supplementary Fig. 7, the first layer film shows isotopic rings, characteristic of surface absorption growth[29]. Based on the distance between isotopic rings, a very high mean growth rate of 133 µm min$^{-1}$ was extracted, indicating the high catalytic activity of liquid $Pt_3Si$. More importantly, the second layer domains always remain AB-stacked with the first layer, even if the orientations of the first layer domain are changed, giving a further evidence of interlayer epitaxy. For further analyses on the formation of the second layer, we selected a specific region, which contains the first layer with the boundary of $^{12}C$ and $^{13}C$ isotopic rings and a second layer domain across the boundary (Fig. 4a, b). Figure 4c shows the Raman spectra taken from the four different positions in Fig. 4a. Note that the position 1 and 2 in the monolayer region show typical Raman spectra of $^{13}C$ and $^{12}C$ MLG, respectively. In contrast, both position 3 and 4 in bilayer region show the Raman spectra of mixed $^{12}C$ and $^{13}C$ isotopes. Since no $^{13}C$ was used in the second step, the much higher intensity of $^{13}C$ peak than $^{12}C$ peak in position 3 and the mixed $^{12}C$ and $^{13}C$ isotopes in position 4 indicate that the second layer was grown epitaxially underneath the first layer through precipitation of the homogeneously mixed $^{12}C$ and $^{13}C$ atoms that were dissolved in the solid Pt core in the first step.

We also performed isotope experiments using one single isotope in each step to understand the formation of the second layer. $^{13}CH_4$ was first supplied to form a MLG film in the first step (10 min), followed by using only $^{12}CH_4$ in the second step with a short cooling time to form isolated second layer domains (60 min). As shown in Supplementary Fig. 9, the bilayer regions are AB-stacked, and both the monolayer and bilayer regions only consist of $^{13}C$, without $^{12}C$ being detected. This gives another evidence that the second layer was grown epitaxially through precipitation of the carbon atoms that were dissolved in the solid Pt core in the first step. Compared to the liquid $Pt_3Si$ above the Pt domains, the liquid $Pt_3Si$ nearby the GBs of Pt can get more carbon supply because of the locally enhanced carbon segregation[30]. As a result, the second layer preferentially nucleates and grows nearby the GBs of Pt in the beginning (Supplementary Fig. 10). The cooling rate of the second step also has an important influence on the graphene growth on $Pt_3Si/Pt$ substrate since it can affect the precipitation of dissolved carbon. As shown in Supplementary Fig. 11, multilayer graphene regions are formed at higher cooling rates. Moreover, both the thickness and coverage of multilayer regions (>2 layers) increase and the coverage of bilayer regions decreases with increasing the cooling rate from 12.5 to 32.5 °C h$^{-1}$. When decreasing the cooling rate from 12.5 to 9.375 °C h$^{-1}$, uniform AB-BLG film is formed as well but the cooling time is extended from 6 to 8 h.

It has been reported that the AB-BLG has stronger van der Waals interaction (42.6 meV per C atom[31]) than the BLG with non-AB stacking (8.4 meV per C atom[32]). Therefore, it is an energy favorable process to form AB stacking if the precipitated bottom graphene nucleus is easily movable on the substrate.

Basically, the movement depends on two factors: one is the binding strength between graphene nucleus and the substrate, and the other is the energy for the movement (including rotation and translation) of graphene nucleus on the substrate. In order to understand the interlayer epitaxial growth mechanism of the bottom layer underneath the pre-formed top layer on liquid $Pt_3Si$ surface, we performed ab initio molecular dynamics (MD) simulations at 1500 K to study the movement of a small graphene nucleus consisting of 54 carbon atoms on $Pt_3Si$ (001). For comparison, we also studied the case on Pt (111), which has relatively low binding energy with graphene among the solid substrates that were used for graphene growth[33]. As shown in Supplementary Fig. 12, at 1500 K, Pt (111) still remains a solid state, while $Pt_3Si$ (001) turns into liquid with Pt and Si atoms taking disordered distribution. DFT calculations show that the binding strength of graphene nucleus with solid Pt (111) and liquid $Pt_3Si$ is very similar (Supplementary Fig. 13), indicating the binding strength is not the reason for the preferential formation of AB-BLG on liquid $Pt_3Si$.

Importantly, it is worth noting that the graphene nucleus takes a flatter morphology on liquid $Pt_3Si$ surface than on Pt (111) surface (Fig. 4d, e). Therefore, the movement of graphene nucleus on liquid $Pt_3Si$ surface should be much easier than on solid Pt (111) surface. Further calculations show that a 1.5 Å translation of graphene nucleus on Pt (111) and liquid $Pt_3Si$ leads to an energy change of 5.75 and 1.47 eV (106.5 and 27.2 meV per C atom), respectively (Supplementary Fig. 14,15). Moreover, the energy changes caused by the rotation of graphene nucleus on Pt (111) are also much larger than on liquid $Pt_3Si$ for different rotation angles (Fig. 4f and Supplementary Figs. 14–16). The highest energy change for the rotation of graphene nucleus on Pt (111) is 4.05 eV with a rotation angle of 27°, while it is only 1.60 eV on liquid $Pt_3Si$ with a rotation angle of 40° (Fig. 4f). Figure 4g–i shows two examples of graphene nucleus movements on liquid $Pt_3Si$ and the corresponding energy requirements. These results confirm that both the rotation and translation of graphene nucleus are energetically easier on liquid $Pt_3Si$ than on solid Pt (111). Therefore, the precipitated bottom graphene can form energy favorable AB stacking with the pre-formed top graphene layer on liquid $Pt_3Si$ during the growth, showing interlayer epitaxial growth behavior as shown in Fig. 2f and Supplementary Fig. 7. Based on the above understandings, we gave the schematic for the interlayer epitaxial growth of AB-BLG film on the liquid $Pt_3Si$/solid Pt substrate in Fig. 5.

## Discussion
Besides AB-BLG, this interlayer epitaxy method provides a general strategy to grow uniform multilayer graphene with a specific stacking order. For instance, improving the precipitation of carbon atoms in the cooling process, by simply decreasing the final temperature from 1025 to 1000 °C, leads to the epitaxial growth of the third layer underneath the AB-BLG film on the same $Pt_3Si$/Pt substrate, forming high-quality ABA-stacked trilayer graphene domains (Supplementary Fig. 17). It is reasonable to believe that wafer-size uniform ABA-stacked trilayer graphene films could be achieved by optimizing the thickness of Pt foil and experimental parameters. Such unique interlayer epitaxy also enables the multilayer graphene regions obtained at a higher cooling rate having specific stacking order (Supplementary Fig. 11). The trilayer regions are ABA-stacked, and the four-layer regions are ABAB-stacked. In principle, this interlayer epitaxy approach is applicable to any metal foils that have high carbon solubility and high melting point, and in particular, can form low-melting-point intermetallic compounds with other elements, such as Pd-Si/Pd, Fe-Si/Fe, Ir-Si/Ir, Pt-P/Pt, and Ir-P/Ir. For example, based on this

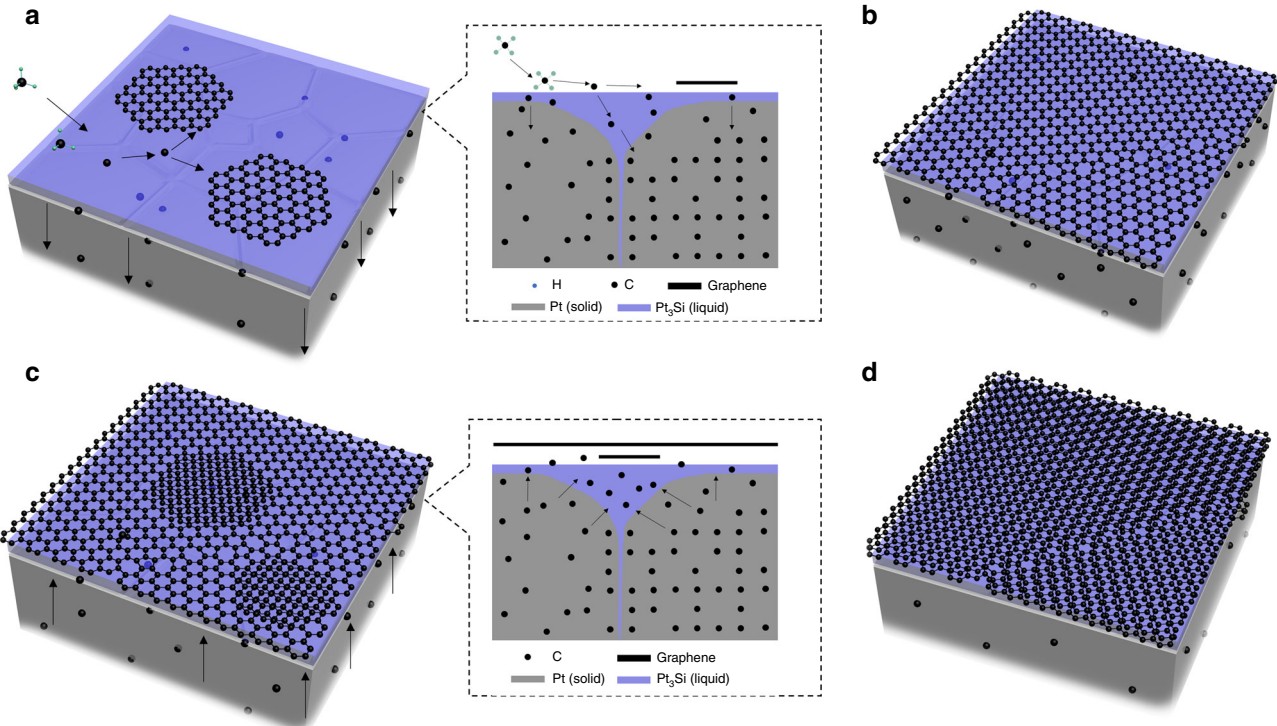

**Fig. 5** Schematic for the interlayer epitaxial growth of AB-BLG film on liquid $Pt_3Si$/solid Pt substrate. **a** The growth of isolated MLG domains in the constant-temperature step. During this process, the excess carbon atoms diffuse through the liquid $Pt_3Si$ shell into the solid Pt core. **b** The formation of a continuous MLG film by merging of the expanded domains. **c** Interlayer epitaxy of isolated second layer domains underneath the pre-formed MLG during slow-cooling process, through the precipitation of the dissolved carbon atoms in the first step. **d** The formation of continuous uniform AB-BLG through the expansion and merging of the second layer domains with continuous supply of the dissolved carbon atoms. Insets in (**a**) and (**c**) are the schematic cross sections. In (**c**) and (**d**), $CH_4$ supply is maintained to prevent $H_2$ etching of the pre-formed MLG

strategy, we also synthesized high-quality AB-BLG films using a liquid Pd silicide/solid Pd substrate (Supplementary Fig. 18), demonstrating the universality of our method. We anticipate that this interlayer epitaxy method can also be used to grow large-area uniform multilayers of other 2D materials, with specific stacking order, which will provide possibilities for investigating interesting properties and intriguing technological applications of multilayer 2D materials.

## Methods

**$Pt_3Si$/Pt substrate preparation.** Pristine Pt substrate (250 μm thick, Alfa, 99.999 wt% metal basis) was first rinsed with DI water, acetone and ethanol in sequence for 1 h each. A 500-nm-thick Si film was then deposited onto the pristine Pt by magnetron sputtering. After that, the Si/Pt substrate was loaded into a boron nitride holder (IMR BN laboratory) inside a fused-silica tube (inner diameter: 22 mm) of a tube furnace (Lindberg Blue M, Thermo Scientific), heated to 1100 °C under the protection of $H_2$ (99.999% purity, 500 sccm), and then annealed for 12 h to obtain $Pt_3Si$/Pt substrate.

**CVD growth of AB-BLG films and single-crystal MLG domains.** A typical procedure for the CVD growth of AB-BLG films on $Pt_3Si$/Pt substrate was shown in Supplementary Fig. 1, which basically includes two steps. After increasing the temperature to 1100 °C under the protection of $H_2$ (500 sccm), $CH_4$ (99.999% purity, 3 sccm) was introduced to grow graphene for 10 min (the first step). Then, the temperature was decreased slowly to 1025 °C with a constant cooling rate of 12.5 °C h$^{-1}$, while keeping the flow of $H_2$ (500 sccm) and $CH_4$ (5 sccm) mixture (the second step). After that, the substrate was quickly pulled out of the high-temperature zone, the furnace was shut down, and the $CH_4$ and $H_2$ flows were turned off when the furnace temperature was lower than 800 °C.

The single-crystal MLG domains for nanoindentation measurements were prepared as reported previously[20]. A piece of Pt foil (250 μm thick, Alfa, 99.999 wt % metal basis, 1 cm × 1 cm) was loaded into a fused-silica tube (inner diameter: 22 mm) and heated to 1000 °C under $H_2$ atmosphere (500 sccm) for 10 min to remove residual carbon or organic substances. Then $CH_4$ (4.5 sccm) was introduced into the tube to initiate graphene growth. After 40 min, we stopped the

feeding of $CH_4$ and rapidly removed the Pt foil from the high-temperature zone to stop graphene growth.

**Electrochemical bubbling transfer of graphene.** The $Pt_3Si$/Pt (or Pt) substrate with the grown graphene was spin-coated with PMMA (950 kDa molecular weight, Sigma, 4.5 wt% in ethyllactate) at 2000 r.p.m for 60 s, and then cured at 110 °C for 10 min. After that, the PMMA/graphene/$Pt_3Si$/Pt (or PMMA/graphene/Pt) stack was dipped in a 1 M NaOH aqueous solution as the cathode with a constant current of 0.2 A, and a Pt foil was used as the anode. After the PMMA/graphene film was separated from the $Pt_3Si$/Pt (or Pt) substrate by $H_2$ bubbles, it was stamped on the target substrate ($SiO_2$/Si, quartz, or TEM grid) and finally acetone was used to remove PMMA at room temperature.

**Structure characterizations.** The structure of $Pt_3Si$/Pt substrates was characterized by optical microscope (Nikon LV100D), SEM (Nova Nano SEM 430, acceleration voltage of 15 kV), X-ray photoelectron spectroscopy (XPS, ESCALAB 250, Al Kα, $5 \times 10^{-10}$ torr) and XRD (Rigaku diffractometer with Cu Kα radiation).

The CVD-grown AB-BLG films were characterized by SEM, optical microscope (Nikon LV100D), UV-vis-NIR spectroscopy (Varain Cary 5000 spectrometer), Raman spectroscopy (JY HR800, 532 nm laser, 1 μm spot size, 5 s integration time, laser power below 2 mW), and TEM (FEI Titan Cube Themis G2 300 equipped with double spherical aberration correctors, 80 kV; FEI Tecnai T12, 120 kV). The mechanically exfoliated MLG and AB-BLG samples on $SiO_2$ (285 nm)/Si substrates were characterized by Raman spectroscopy and AFM (Multimode 8, Bruker). The CVD-grown single-crystal MLG domains were characterized by optical microscopy and Raman spectroscopy.

**Mechanical property measurements.** The force curve measurements and data processing of CVD-grown AB-BLG and single-crystal MLG domains were the same as those reported previously[24,25]. In general, AB-BLG films and single-crystal MLG domains were first transferred onto a $SiO_2$ (285 nm thick)/Si substrate with circular wells (diameter 1 μm, depth 300 nm) by electrochemical bubbling to obtain suspended membrane. The force curves were then measured by AFM (Multimode 8, Bruker) with a diamond tip (AD-150-NM, Adama). After the topography image was obtained, we changed the sample's XY position to locate the membrane's center right beneath the tip and performed nanoindentation measurement. The cantilever spring constant was 172.8 N m$^{-1}$, which was calibrated by the Sader

method[34]. The radius of curvature of tip was 11.07 nm. During an indentation, the sample moved up vertically with a constant rate of 50 nm s$^{-1}$ to make contact with the tip and deflect the membrane according to the presetting upper/lower limits of z-piezo extension. In this procedure, only the change of the z-piezo extension and the change of cantilever deflection could be measured. The measured force was calculated by multiplying the cantilever deflection by the spring constant. The membrane deflection in **z** direction ($Z$) was further obtained through subtracting the cantilever deflection from the z-piezo extension. A force curve was finally acquired. After the tip made contact with the membrane, the displacement of the membrane in the middle could be acquired from the change of $Z$ value. By adopting a non-linear model, the force-displacement behavior can be approximated[24] as

$$F = \sigma_0^{2D} \pi d + E^{2D} q^3 \frac{d^3}{R^2} \qquad (1)$$

where $\sigma_0^{2D}$ is the pretension accumulated in the membrane, $E^{2D}$ is the elastic constant with unit of force/length, $R$ is the radius of the patterned holes (shown in Fig. 3a), and $q$ is a constant related to Poisson's ratio ($v$) of the membrane, following $q = 1/(1.05-0.15v-0.16v^2)$. We took $v = 0.165$, such that $q = 0.98$[24].

The fracture strength of graphene was measured by loading the membranes to the breaking point. For MLG, it can be modeled in the elastic regime to yield an analytical expression for the breaking strain $\sigma_m$ as a function of breaking force $F$ and tip radius $r$:

$$\sigma_m = \left(\frac{FE}{4\pi r}\right)^{\frac{1}{2}} \qquad (2)$$

Because the slip between the two layers of BLG[25], it is difficult to find a correct model to extract the fracture strength. Since the fracture strength of BLG varies approximately with $r^{1/2}$, we used the measured strengths of MLG, together with the tip radius used in each experiment, to scale the bilayer results as reported in ref. [25].

**Fabrication and measurements of dual-gate AB-BLG FETs.** First, we used a conventional lithography method to fabricate source/drain electrodes of Ti/Au (5 nm/50 nm) by electron beam evaporation on SiO$_2$ (290 nm)/Si substrate. Then, the AB-BLG film was transferred onto the SiO$_2$/Si substrate with source/drain electrodes by using electrochemical bubbling[20]. After 30 nm Al$_2$O$_3$ was deposited by atomic layer deposition (ALD) as the top-gate dielectric, the AB-BLG film was patterned with lithography, wet etching (Tetramethylammonium Hydroxide) and oxygen plasma etching. In order to avoid current leakage, we deposited another 30 nm Al$_2$O$_3$. Finally, the top gate was patterned and deposited with Ti/Au (5 nm/100 nm) by electron beam evaporation. The FET devices were measured using semiconductor analyzer (Agilent B1500A) and probe station (Cascade M150) at room temperature.

**Calculations.** All the calculations were carried out with the Vienna ab initio simulation package[35] within the framework of DFT. The ab initio MD simulation was conducted at 1500 K (about 100 K higher than the experimental temperature) with a 1.0 fs movement for each step in the canonical ensemble employing Nosé-Hoover thermostats (NVT)[36]. The simulation was performed for 8 ps (totally 8000 time steps) with the Gamma-point only. The exchange-correlation effects were treated within the generalized gradient approximation using the Perdew-Burke-Ernzerhof functional[37]. Electron-ion interactions were described using the frozencore projector augmented wave approach[38,39] and the Kohn-Sham one-electron valence states were expanded on the basis of plane waves with a cutoff energy of 300 eV. The system of graphene nucleus on Pt (111) surface, namely graphene (Gr)/Pt (111) for short, was constructed by three Pt (111) atomic layers (168 Pt atoms) and a graphene nucleus consisting 19 C-six member rings (54 C atoms). The system of graphene on Pt$_3$Si (001) surface, namely Gr/Pt$_3$Si (001) for short, was constructed by three Pt$_3$Si (001) atomic layers (64 Si and 192 Pt atoms) and a graphene nucleus consisting 19 C-six member rings (54 C atoms). During the ab initio MD simulations, the bottom Pt atomic layer was fixed for both cases. A large vacuum thickness of 20 Å was used to avoid the periodic interaction at the **z** direction and provide sufficient space for possible structure changes of Gr/Pt (111) and Gr/Pt$_3$Si (001) during the ab initio MD simulation. The binding energies ($E_b$) were calculated by following equation:

$$E_b = E_{total}(Gr) + E_{total}(sub) - E_{total}(Gr/sub) \qquad (3)$$

Here, $E_{total}$ (Gr), $E_{total}$ (sub), and $E_{total}$ (Gr/sub) are the total energies of graphene nucleus, total energies of Pt (111) or Pt$_3$Si (001), and total energies of Gr/Pt (111) or Gr/ Pt$_3$Si (001) without geometry relaxation, respectively. According to this definition, larger $E_b$ means stronger binding between graphene nucleus with substrate.

The relative energies were calculated by following equation:

$$E_r = E_{Gr/sub}(\text{rotation/shift}) - E_{Gr/sub} \qquad (4)$$

Here, $E_{Gr/sub}$ is the total energies of Gr/Pt (111) or Gr/Pt$_3$Si (001) after geometry relaxation, and $E_{Gr/sub}$ (rotation/shift) is the total energies of Gr/Pt (111) or Gr/Pt$_3$Si (001) with different rotation or translation movement after geometry relaxation.

## Data availability

The data that support the findings of this study are available from the corresponding author upon request.

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

## Acknowledgements

We thank Shun Feng for kind help on electrical property measurements. This work was financially supported by the National Key R&D Program of China (No. 2016YFA0200101), National Science Foundation of China (Nos. 51325205, 51290273, 51521091, and 51472249), the Strategic Priority Research Program of Chinese Academy of Sciences (No. XDB30000000), LiaoNing Revitalization Talents Program (No. XLYC1808013), the Program for Guangdong Introducing Innovative and Enterpreneurial Teams, and the Development and Reform Commission of Shenzhen Municipality for the development of the "Low-Dimensional Materials and Devices" discipline. The theoretical calculations in this work were performed on TianHe-1(A) at National Supercomputer Center in Tianjing and Tianhe-2 at National Supercomputer Center in Guangzhou.

## Author contributions

W.R. conceived and supervised the project; W.M. and W.R. designed the experiments; W.M. performed the graphene growth on $Pt_3Si/Pt$ substrate, transfer, structure characterizations and mechanical property measurements; H.L. performed the graphene growth on $Pd_5Si/Pd$ substrate and transfer. M.C. fabricated and measured the FET devices under the supervision of D.S.; L.Y. conducted the theoretical calculations; Z.L. performed the TEM measurements; C.X. and X.X. helped with graphene growth and transfer; W.M. and W.R. analyzed the experimental data; W.M., L.Y., H.C., and W.R. wrote the manuscript. All the authors discussed the results and commented on the manuscript.

## Additional information

**Competing interests:** The authors declare no competing interests.

