## [Peer Review File · Nature Communications]

Reviewers' comments:

Reviewer #1 (Remarks to the Author):

AB-stacked bilayer graphene (AB-BLG) has many unique physical properties and holds great potential for electronic and photonic device applications, but the growth of large-area uniform AB-BLG films is an open challenge. This paper, for the first time, reports the growth of wafer-scale 100% AB-BLG films by using a well-designed core-shell structured liquid/solid substrate, which show superior quality, mechanical and electrical properties comparable to the mechanically exfoliated AB-BLG. Furthermore, a new interlayer epitaxy mechanism was proposed based on systematic isotope labeling experiments and theoretical calculations. More impressively, the interlayer epitaxy was demonstrated to be a universal method by growing ABA-stacked trilayer graphene and AB-BLG on other liquid/solid substrates.

Overall, this is a very interesting and urgent important work with high novelty, which provides a possible general strategy to grow 2D materials with specific stacking order, and the results are well organized and clearly described. I believe it will attract the immediate interest from a broad materials research community. Therefore, I would like to recommend its publication of this paper in Nature Communications.

Two small suggestions:

- 1) The AFM nanoindentation experiments of 2D materials can be influenced by many factors. To make the mechanical characterization of AB-BLG more reliable, I suggest the authors to give the nanoindentation results of monolayer graphene prepared by mechanical exfoliation or CVD measured by the same method and compare these results with those reported in the literature.
- 2) The authors mentioned that the interlayer epitaxy approach might be applicable to the metals that have high carbon solubility, high melting point, and the capability to form low melting point intermetallic compounds with other elements. They demonstrated the universality of this method by growing AB-BLG films on a liquid Pd silicide/solid Pd substrate. It would further improve the quality and impact of the work if the authors could give/predict a few more possible substrate systems that have a potential for achieving interlayer epitaxy of graphene.

Reviewer #2 (Remarks to the Author):

Re: Manuscript#: NCOMMS-19-05917-T

This is a very good work for bi-layer Graphene (BLG) growth using CVD method, which potentially for promising next generation electronic, photonic and spintronic devices. However, I has few concerns on this works before I can make a clear conclusion. Hope authors can help to reply my questions shown below.

- 1) The electronic property of the BLG is key for possible digital devices, and high on/off ratio is essential to open up the potential applications of the digital IC. Please help to provide the on/off ratio of your FET device fabricated using the CVD BLG.
- 2) Page2, line3, Authors reported tunable bandgap $>26\text{meV}$ at 1 v/nm , which is only 25% compared to ref.11. Any possible reason?
- 3) From Figure S1, after 10 min growth of mono-layer graphene at 1000C , authors slowly cool down the CVD temperature with 12.5C/h . If this cooling rate higher or lower, what's the impact on the BLG growth? Authors may has some more room to further improve current CVD methods.
- 4) Page10, line3. Authors claim "we have synthesized AB-BLG with millimeter-sized grains by using a small flow rate ratio of methane to hydrogen in the first step (Fig. 1e and Supplementary Fig. 6)." However, the Fig.1e showed that BLG single domain size is not above $400\mu\text{m}$, how to get the millimeter sized grain?
- 5) Authors keeps claim the Pt3Si/Pt core-shell structures, which is very confused concept. As authors description, the solid Pt is core and liquid Pt3Si is the shell. Could authors provide some cartoon to show how the core-shell structures looks like?
- 6) Page5 line12. Authors claim that Pt3Si shell with low carbon solubility compared to Pt core, can

authors provide the number of the carbon solubility of Pt₃Si?

Reviewer #3 (Remarks to the Author):

The authors report the growth of wafer-scale AB-stacked bilayer graphene film using core-shell structured liquid Pt₃Si/solid Pt substrate. They claimed their grown film is continuous and high-quality based on the electrical and mechanical characterization. They explained the growth of AB-stacked BLG is possible due to different carbon solubility of solid Pt and liquid phase Pt₃Si. The paper was written clearly, and the authors did extensive characterization to prove the quality of their film. I have several questions/comments as below.

1. The authors used the term "100%" even at the title however I'd like to recommend to use different expressions instead of "100%". I think 100% is not a scientifically exact term. I understand the authors want to emphasize the coverage of the film. However, there are several ambiguities such as reference area (wafer, a field of view of the optical image...) and scale of coverage (um- or nm-size holes). I think the continuous polycrystalline film should be a better expression for your sample.
2. For the indentation experiment, are the samples with grain boundaries included?
3. For the isotope experiment in Figure 4a-c, I think it will be more straightforward if you can grow each layer with a single isotope (1st->C¹², 2nd->C¹³ or vice versa).
4. It will be more helpful for readers to understand if you can include a schematic for the growth mechanism and process. Current figures only show atomic behavior at each Pt and Pt₃Si. Showing the whole growth process of AB-stacked BLG on Pt/Pt₃Si substrate as a schematic will be very effective.
5. Is there any data showing the correlation between domain boundaries of Pt and the growth of the 2nd layer? If I understand correctly since the formation of Pt₃Si will be preferentially at the grain boundary, there should be a correlation between domain boundaries of Pt and the growth of the 2nd layer. Optical or Raman images on the growth substrate should show the aforementioned trend if there is any.

Response to reviewers' comments

Reviewer #1 (Remarks to the Author):

AB-stacked bilayer graphene (AB-BLG) has many unique physical properties and holds great potential for electronic and photonic device applications, but the growth of large-area uniform AB-BLG films is an open challenge. This paper, for the first time, reports the growth of wafer-scale 100% AB-BLG films by using a well-designed core-shell structured liquid/solid substrate, which show superior quality, mechanical and electrical properties comparable to the mechanically exfoliated AB-BLG. Furthermore, a new interlayer epitaxy mechanism was proposed based on systematic isotope labeling experiments and theoretical calculations. More impressively, the interlayer epitaxy was demonstrated to be a universal method by growing ABA-stacked trilayer graphene and AB-BLG on other liquid/solid substrates.

Overall, this is a very interesting and urgent important work with high novelty, which provides a possible general strategy to grow 2D materials with specific stacking order, and the results are well organized and clearly described. I believe it will attract the immediate interest from a broad materials research community. Therefore, I would like to recommend its publication of this paper in Nature Communications.

Reply: We thank the reviewer very much for positive comments.

1) The AFM nanoindentation experiments of 2D materials can be influenced by many factors. To make the mechanical characterization of AB-BLG more reliable, I suggest the authors to give the nanoindentation results of monolayer graphene prepared by mechanical exfoliation or CVD measured by the same method and compare these results with those reported in the literature.

Reply: We thank the reviewer very much for kind suggestions.

According to the reviewer's suggestion, we have performed nanoindentation measurements on the CVD-grown high-quality single-crystal monolayer graphene (MLG) with the same method. Single-crystal MLG domains of hundreds of

micrometers were first transferred onto a SiO₂ (285 nm thick)/Si substrate with circular wells (diameter 1 μm, depth 300 nm) to obtain suspended membranes (Fig. R1a). The absence of defect-related D peak in Raman spectrum indicates the high quality of MLG (Fig. R1b). Analyzing all the measured data using Eq. S1 and Eq. S2 yields an average 2D Young's modulus of ~334.02 N m⁻¹ and breaking strength of ~55.31 N m⁻¹ (Fig. R1c,d). These results are well consistent with the previous reports of CVD-grown large-grain MLG (*Science* **340**, 1073-1076, 2013), confirming the reliability of the AFM nanoindentation measurements in this work.

We have added the figure and related discussions in the revised manuscript.

Figure R1 | Mechanical properties of the CVD-grown single-crystal MLG domains. **a,b**, Optical image (**a**) and typical Raman spectrum (**b**) of the CVD-grown single-crystal MLG transferred onto a SiO₂/Si substrate with an array of holes with 1-μm diameter. **c,d**, Histograms of elastic stiffness (**c**) and breaking strength (**d**). Dashed lines represent Gaussian fits to the data.

2) The authors mentioned that the interlayer epitaxy approach might be applicable to the metals that have high carbon solubility, high melting point, and the capability to

form low melting point intermetallic compounds with other elements. They demonstrated the universality of this method by growing AB-BLG films on a liquid Pd silicide/solid Pd substrate. It would further improve the quality and impact of the work if the authors could give/predict a few more possible substrate systems that have a potential for achieving interlayer epitaxy of graphene.

Reply: We thank the reviewer very much for constructive suggestion.

We analyzed the phase diagrams and found that the following substrate systems can form a core (solid)-shell (liquid) structure which may achieve interlayer epitaxy of graphene, Fe-Si/Fe, Ir-Si/Ir, Pt-P/Pt, and Ir-P/Ir.

We have added the above substrate systems in the revised manuscript.

Reviewer #2 (Remarks to the Author):

This is a very good work for bi-layer Graphene (BLG) growth using CVD method, which potentially for promising next generation electronic, photonic and spintronic devices. However, I has few concerns on this works before I can make a clear conclusion. Hope authors can help to reply my questions shown below.

Reply: We thank the reviewer very much for positive comments.

1) The electronic property of the BLG is key for possible digital devices, and high on/off ratio is essential to open up the potential applications of the digital IC. Please help to provide the on/off ratio of your FET device fabricated using the CVD BLG.

Reply: We thank the reviewer very much for kind comment.

The on/off ratio of the BLG dual-gate FET device in this work is ~ 10 at $D \approx 1.0 \text{ V nm}^{-1}$, which is comparable to that of exfoliated AB-BLG (9.28 - 12.5) with similar device structure at the same electrical displacement. (*Nano Lett.* **10**, 715-718, 2010).

The on/off ratio is positively correlation with the bandgap within certain range. In

this work, the top-gate dielectrics of the BLG dual-gate FET device (Al_2O_3) were directly deposited by atomic layer deposition (ALD). The inert surface of our high-quality BLG film leads to the lack of reactive surface sites for the initiation of ALD growth (*Adv. Mater. Interfaces* **4**, 1700232, 2017). Therefore, the top-gate dielectrics of Al_2O_3 in this work are not very uniform to sustain higher electric field to open the bandgap of BLG further, which limits the on/off ratio. We believe that the on/off ratio of our BLG film can be further improved by using an organic seed layer to facilitate deposition of dielectrics (*Nano Lett.* **10**, 715-718, 2010) or using h-BN as back and top-gate dielectrics (*Nat. Nanotechnol.* **11**, 426-431, 2016).

We have added the number of on/off ratio and related discussions in the revised manuscript.

2) Page2, line3, Authors reported tunable bandgap >26 meV at 1 V/nm, which is only 25% compared to ref.11. Any possible reason?

Reply: We thank the reviewer very much for kind comment.

The AB-BLG in Ref. 11 was encapsulated between two h-BN flakes, which act as back and top-gate dielectrics. In contrast, the AB-BLG in this work was encapsulated between SiO_2 and Al_2O_3 . Compared to SiO_2 and Al_2O_3 , h-BN has an atomically smooth surface that is relatively free of dangling bonds and charge traps. Moreover, it has a lattice constant similar to that of graphite, and have large optical phonon modes and a large electrical bandgap. Therefore, the graphene on h-BN can maintain its intrinsic properties. For example, it has been reported that the graphene devices on h-BN have mobilities and carrier inhomogeneities that are almost an order of magnitude better than devices on SiO_2 (*Nat. Nanotechnol.* **5**, 722-726, 2010). We suggest that the use of h-BN as dielectrics plays an important role in achieving a more efficient bandgap tuning as shown in Ref. 11.

We have added related discussions in the revised manuscript.

3) From Figure S1, after 10 min growth of mono-layer graphene at 1100 °C, authors slowly cool down the CVD temperature with 12.5C/h. If this cooling rate higher or lower, what's the impact on the BLG growth? Authors may have some more room to further improve current CVD methods.

Reply: We thank the reviewer very much for constructive comments.

12.5 °C/h is the optimized cooling rate for the growth of uniform AB-BLG in our experiments. We have studied the growth of BLG with different cooling rates in the cooling step, and the results were shown in Fig. R2. It can be seen that multilayer graphene regions were formed at higher cooling rates (Fig. R2a-d). Moreover, the thickness and coverage of the multilayer regions (>2 layers) increased and the coverage of bilayer regions decreased with increasing the cooling rate from 12.5 to 32.5 °C/h. In the cooling step, the carbon solubility in the Pt core was decreased with decreasing the temperature (*Materials Park Ohio*, 2705-2708, 1990), leading to the precipitation of carbon atoms. When the cooling rate is relatively high, the concentration of precipitated carbon atoms may be higher than the critical nucleation concentration, especially nearby the Pt domain boundaries. As a result, multilayer graphene domains were formed instead of growing up of BLG (Fig. R2a-d). With a cooling rate of 12.5 °C/h, uniform AB-BLG film was obtained (Fig. R2e,f). When decreasing the cooling rate from 12.5 to 9.375 °C/h, uniform AB-BLG film was formed as well (Fig. R2g,h) but the cooling time was extended from 6 h to 8 h. Therefore, we used a cooling rate of 12.5 °C/h in our experiments. It is worth noting that all the bilayer regions (2D peak FWHM: 52.85 – 56.23 cm⁻¹) obtained at different cooling rates are AB-stacked, trilayer regions (2D peak FWHM: 61.65 – 64.59 cm⁻¹) are ABA-stacked, and four-layer regions (2D peak FWHM: 64.59 – 67.53 cm⁻¹) are ABAB-stacked (*Nano Lett.* **11**, 164-169, 2011), which further confirm that interlayer epitaxy is a universal growth behavior of graphene on liquid Pt₃Si/solid Pt substrate.

We have added these figures and related discussions in the revised manuscript.

Figure R2 | Graphene synthesized with different cooling rates in the second step.
a,c,e,g, Optical images of graphene transferred on SiO₂/Si substrates. The cooling rates are 32.5 °C/h (a), 18.75 °C/h (c), 12.5 °C/h (e), and 9.375 °C/h (g) in the cooling

step, respectively. $T_1 = 1100\text{ }^\circ\text{C}$; $T_2 = 1025\text{ }^\circ\text{C}$; $t_1 = 10\text{ min}$. **b,d,f,g**, Raman spectra taken for the regions marked in **(a)**, **(c)**, **(e)** and **(g)**, respectively.

4) Page10, line3. Authors claim “we have synthesized AB-BLG with millimeter-sized grains by using a small flow rate ratio of methane to hydrogen in the first step (Fig. 1e and Supplementary Fig. 6).” However, the Fig.1e showed that BLG single domain size is not above $400\text{ }\mu\text{m}$, how to get the millimeter sized grain?

Reply: We thank the reviewer very much for kind comments.

Fig.1e shows an intermediate stage of the growth process of uniform continuous AB-BLG film. It can be seen that there is still a few hundreds of micrometer space for further expansion of the single domain in the following growth process. As shown in the manuscript, the 2nd layer of BLG is grown epitaxially underneath the 1st layer (Fig. 2f and Supplementary Fig. 7). Therefore, the grain size and grain boundaries of the AB-BLG film are determined by the grain structure of the pre-formed 1st layer graphene. Supplementary Fig. 7 shows another intermediate stage of the growth process under the same growth conditions. The isotopic rings of the 1st layer indicate that the grain size of the 1st layer can reach millimeter size, which was synthesized by using a small flow rate ratio of methane to hydrogen in the first step. Therefore, we suggest that the grain size of finally formed continuous AB-BLG film (Fig. 1f) with the same conditions is on the order of millimeter.

We have added related discussions in the revised manuscript.

5) Authors keeps claim the $\text{Pt}_3\text{Si}/\text{Pt}$ core-shell structures, which is very confused concept. As authors description, the solid Pt is core and liquid Pt_3Si is the shell. Could authors provide some cartoon to show how the core-shell structures looks like?

Reply: We thank the reviewer very much for constructive suggestion.

We have provided a cartoon to show the shell-core structure of $\text{Pt}_3\text{Si}/\text{Pt}$ substrate and the interlayer epitaxial growth process of AB-BLG film on such substrate in the revised manuscript, as shown in Fig. R3. In addition, in order to avoid confusing, we have changed “ $\text{Pt}_3\text{Si}/\text{Pt}$ core-shell structures” to “ $\text{Pt}_3\text{Si}/\text{Pt}$ shell-core structures”.

Figure R3 | Schematic for the interlayer epitaxial growth of AB-BLG film on liquid Pt₃Si/solid Pt substrate. **a**, The growth of isolated monolayer graphene domains in the constant-temperature step. During this process, the excess carbon atoms diffuse through the liquid Pt₃Si shell into the solid Pt core. **b**, The formation of a continuous monolayer graphene film by merging of the expanded domains. **c**, Interlayer epitaxy of isolated 2nd layer domains underneath the pre-formed monolayer graphene during slow-cooling process, through the precipitation of the dissolved carbon atoms in the first step. **d**, The formation of continuous uniform AB-BLG through the expansion and merging of the 2nd domains with continuous supply of the dissolved carbon atoms. Insets in (a) and (b) are the schematic cross sections. In (c) and (d), CH₄ supply is maintained, which is mainly to prevent H₂ etching of the pre-formed monolayer graphene.

6) Page5 line12. Authors claim that Pt₃Si shell with low carbon solubility compared to Pt core, can authors provide the number of the carbon solubility of Pt₃Si?

Reply: We thank the reviewer very much for kind suggestion.

We have checked the ternary phase diagram of Pt, C, and Si, and only two phase diagrams at the temperatures of 298 K and 1023 K are available, both of which show no ternary phase of Pt, C, and Si

(https://materials.springer.com/isp/phase-diagram/docs/c_0955481; https://materials.springer.com/isp/phase-diagram/docs/c_0200741; *J. Am. Ceram. Soc* **45**, 268-273, 1962). These indicate that the carbon solubility of Pt₃Si is almost zero at these two temperatures.

In order to identify the carbon solubility of liquid Pt₃Si at the growth temperature of graphene, as shown in our manuscript, we performed an experiment as follows. The Pt₃Si/Pt substrate was quickly pulled out of the high-temperature zone after monolayer graphene was grown on it at 1100 °C in 10 min. Such operation ensures a rapid quenching of the reactions and allows the carbon distribution in the substrate during growth to be captured. Then we used XPS to analyze the substrate obtained, and results were shown in Supplementary Fig. 2. Because of the presence of monolayer graphene, strong C1s peak was observed in the substrate without Ar⁺ etching. After 60 and 120 s etching, the Pt₃Si was exposed and showed a very low C1s peak, indicating that the liquid Pt₃Si has a low carbon solubility. The formation of uniform monolayer graphene on Pt₃Si gives another evidence of the low carbon solubility of Pt₃Si (Supplementary Fig. 3). However, we could not give the exact number of the carbon solubility of Pt₃Si.

Reviewer #3 (Remarks to the Author):

The authors report the growth of wafer-scale AB-stacked bilayer graphene film using core-shell structured liquid Pt₃Si/solid Pt substrate. They claimed their grown film is continuous and high-quality based on the electrical and mechanical characterization. They explained the growth of AB-stacked BLG is possible due to different carbon solubility of solid Pt and liquid phase Pt₃Si. The paper was written clearly, and the authors did extensive characterization to prove the quality of their film. I have several questions/comments as below.

Reply: We thank the reviewer very much for positive comments.

1) The authors used the term "100%" even at the title however I'd like to recommend

to use different expressions instead of "100%". I think 100% is not a scientifically exact term. I understand the authors want to emphasize the coverage of the film. However, there are several ambiguities such as reference area (wafer, a field of view of the optical image...) and scale of coverage (um- or nm-size holes). I think the continuous polycrystalline film should be a better expression for your sample.

Reply: We thank the reviewer very much for kind suggestion. According to the reviewer's suggestion, we have changed the related expressions to "continuous uniform AB-BLG film".

2) For the indentation experiment, are the samples with grain boundaries included?

Reply: We thank the reviewer very much for kind comments.

As shown in the manuscript, the grain size of the AB-BLG film obtained is on the order of millimeter (Fig. 1e and Supplementary Fig. 7). As for the indentation experiments, the graphene was suspended over the holes of 1 μm in diameter in a SiO_2/Si substrate (Fig. 3a inset). That is, the indentation area is 1 μm . Therefore, the probability of containing grain boundaries in the indentation area should be very low. In addition, the values of Young's modulus and breaking strength were obtained based on the statistical results of dozens of samples. The influence of the grain boundaries on the mechanical properties of AB-BLG film can be ignored in our experiments, but it is a very interesting topic deserving further studies in the future.

We have added related discussions in the revised manuscript.

3) For the isotope experiment in Figure 4a-c, I think it will be more straightforward if you can grow each layer with a single isotope (1st->C12, 2nd->C13 or vice versa).

Reply: We thank the reviewer very much for constructive suggestion.

As the reviewer mentioned, it is surely more straightforward to grow each layer with a single isotope to prove that the 2nd layer is grown underneath the 1st layer through precipitation of carbon atoms. However, we cannot obtain such samples because the carbon source for the growth of 2nd layer is the same as that for the

growth of 1st layer. For example, ¹³CH₄ was first supplied to form a monolayer graphene film in the first step (10 min), followed by using only ¹²CH₄ in the second step with a shorter cooling time to form isolated 2nd layer domains (60 min). As shown in Fig. R4, the bilayer regions are AB-stacked. More importantly, both the monolayer and bilayer regions only consist of ¹³C, without ¹²C being detected. This gives direct evidence that the 2nd layer was grown epitaxially underneath the 1st layer through precipitation of ¹³C atoms that were dissolved in the solid Pt in the first step.

Note that the isotope experiments presented in the manuscript can give more details of the growth process, such as the growth rate and mechanism of the 1st layer, the grain size of the AB-BLG film, and the stacking order nearby the grain boundaries of the 1st layer. Therefore, we added the new isotope experiment results (Fig. R4) in the supplementary information and the related discussions in the main text.

Figure R4 | Isotope experiment of growing AB-BLG film on the Pt₃Si/Pt substrate with a single isotope. a-d, Optical image (a), Raman spectra (b) taken from the two regions marked in (a), and Raman maps of 2D-peak FWHM (52.1 – 56.3 cm⁻¹) (c) and position (2590 – 2600 cm⁻¹) (d) of the CVD-grown AB-BLG

transferred on SiO₂/Si substrate. ¹³CH₄ was only supplied in the first step (in total 10 min), followed by using only ¹²CH₄ in the second step for 60 min.

4) It will be more helpful for readers to understand if you can include a schematic for the growth mechanism and process. Current figures only show atomic behavior at each Pt and Pt₃Si. Showing the whole growth process of AB-stacked BLG on Pt/Pt₃Si substrate as a schematic will be very effective.

Reply: We thank the reviewer very much for the constructive suggestion.

According to the reviewer's suggestion, we have added a schematic to show the whole growth process and growth mechanism of AB-BLG on liquid Pt₃Si/solid Pt substrate, as shown in Fig. R3.

Figure R3 | Schematic for the interlayer epitaxial growth of AB-BLG film on liquid Pt₃Si/solid Pt substrate. a, The growth of isolated monolayer graphene domains in the constant-temperature step. During this process, the excess carbon atoms diffuse through the liquid Pt₃Si shell into the solid Pt core. **b,** The formation of a continuous monolayer graphene film by merging of the expanded domains. **c,** Interlayer epitaxy of isolated 2nd layer domains underneath the pre-formed monolayer graphene during slow-cooling process, through the precipitation of the dissolved

carbon atoms in the first step. **d**, The formation of continuous uniform AB-BLG through the expansion and merging of the 2nd domains with continuous supply of the dissolved carbon atoms. Insets in **(a)** and **(b)** are the schematic cross sections. In **(c)** and **(d)**, CH₄ supply is maintained, which is mainly to prevent H₂ etching of the pre-formed monolayer graphene.

5) Is there any data showing the correlation between domain boundaries of Pt and the growth of the 2nd layer? If I understand correctly since the formation of Pt₃Si will be preferentially at the grain boundary, there should be a correlation between domain boundaries of Pt and the growth of the 2nd layer. Optical or Raman images on the growth substrate should show the aforementioned trend if there is any.

Reply: We thank the reviewer very much for constructive suggestion.

Due to the poor optical contrast and negligible Raman intensity of BLG on Pt₃Si/Pt substrate, it is very hard to provide valuable information of graphene growth on Pt₃Si/Pt substrate by optical microscopy and Raman spectroscopy. Instead, we used SEM to study the correlation between domain boundaries of Pt and the growth of the 2nd layer. Fig. R5 shows BLG domains that were synthesized by intentionally reducing the cooling time ($T_1 = 1100$ °C; $T_2 = 1075$ °C; $t_1 = 10$ min; $t_2 = 120$ min). It can be seen that the 2nd layer preferentially nucleated and grew nearby the domain boundaries of Pt in the beginning. This is because the higher carbon supply at the liquid Pt₃Si nearby the domain boundaries of Pt, which was caused by the locally enhanced carbon segregation (*Nano Lett.* **11**, 2628-2633, 2011).

We have added this figure and the related discussions in the revised manuscript.

Figure R5 | SEM images of bilayer graphene domains grown on Pt₃Si/Pt substrate. T₁ = 1100 °C; T₂ = 1075 °C; t₁ = 10 min; t₂ = 120 min.

REVIEWERS' COMMENTS:

Reviewer #1 (Remarks to the Author):

I have carefully reviewed the author's manuscript and response to the reviewers. The authors did excellent work on AB-stacked bilayer graphene films. I would like to recommend the publication of this paper in Nature Communications.

Reviewer #2 (Remarks to the Author):

Authors successfully address most of my questions, and give a proper feedback in the right shape. I would like thanks first for all the high quality work done for the revision version, and agree to accept this work to be published. However, it will be better if several minor questions to be considered.

1) One of the key to use this large scale ABG film is the transfer method from CVD substrate to any other SiO_x or BN substrate. Authors claim to use the electrochemical bubbling method, please release more details if possible.

2) For Figure 5, on initial growth stage, graphen nucleation should prefer to start from grain boundary of the liquid Pt₃Si/solid Pt substrate. Please help to transfer this information on Fig.5 if possible.

Reviewer #3 (Remarks to the Author):

The authors have revised the manuscript based on the reviewer's comments reasonably. I recommend publishing the manuscript to Nature Communications.

Response to reviewers' comments

Reviewer #1 (Remarks to the Author):

I have carefully reviewed the author's manuscript and response to the reviewers. The authors did excellent work on AB-stacked bilayer graphene films. I would like to recommend the publication of this paper in Nature Communications.

Reply: We thank the reviewer very much for positive comments.

Reviewer #2 (Remarks to the Author):

Authors successfully address most of my questions, and give a proper feedback in the right shape. I would like thanks first for all the high-quality work done for the revision version, and agree to accept this work to be published. However, it will be better if several minor questions to be considered.

Reply: We thank the reviewer very much for positive comments.

1. One of the keys to use this large scale ABG film is the transfer method from CVD substrate to any other SiO_x or BN substrate. Authors claim to use the electrochemical bubbling method, please release more details if possible.

Reply: According to the reviewer's suggestion, we have provided the details of the electrochemical bubbling transfer in the revised manuscript.

2. For Figure 5, on initial growth stage, graphene nucleation should prefer to start from grain boundary of the liquid Pt_3Si /solid Pt substrate. Please help to transfer this information on Fig.5 if possible.

Reply: We thank the reviewer very much for kind suggestions.

As shown in Fig. R1, we have adjusted the details of the inset of Figure. 5c to show the locally enhanced carbon segregation and preferential nucleation of the graphene layer nearby the domain boundaries of Pt on the initial growth stage.

We have replaced Figure 5 with this figure in the revised manuscript.

Figure R1 | Schematic for the interlayer epitaxial growth of AB-BLG film on liquid Pt₃Si/solid Pt substrate. **a**, The growth of isolated monolayer graphene domains in the constant-temperature step. During this process, the excess carbon atoms diffuse through the liquid Pt₃Si shell into the solid Pt core. **b**, The formation of a continuous monolayer graphene film by merging of the expanded domains. **c**, Interlayer epitaxy of isolated 2nd layer domains underneath the pre-formed monolayer graphene during slow-cooling process, through the precipitation of the dissolved carbon atoms in the first step. **d**, The formation of continuous uniform AB-BLG film through the expansion and merging of the 2nd domains with continuous supply of the dissolved carbon atoms. Insets in (a) and (c) are the schematic cross sections. In (c) and (d), CH₄ supply is maintained to prevent H₂ etching of the pre-formed monolayer graphene.

Reviewer #3 (Remarks to the Author):

The authors have revised the manuscript based on the reviewer's comments reasonably. I recommend publishing the manuscript to Nature Communications.

Reply: We thank the reviewer very much for positive comments.